# The Role of Epigenomics in Osteoporosis and Osteoporotic Vertebral Fracture

**DOI:** 10.3390/ijms21249455

**Published:** 2020-12-11

**Authors:** Kyoung-Tae Kim, Young-Seok Lee, Inbo Han

**Affiliations:** 1Department of Neurosurgery, School of Medicine, Kyungpook National University, Daegu 41944, Korea; nskimkt7@gmail.com (K.-T.K.); leeys1026@hanmail.net (Y.-S.L.); 2Department of Neurosurgery, Kyungpook National University Hospital, Daegu 41944, Korea; 3Department of Neurosurgery, Kyungpook National University Chilgok Hospital, Daegu 41944, Korea; 4Department of Neurosurgery, CHA University School of medicine, CHA Bundang Medical Center, Seongnam-si, Gyeonggi-do 13496, Korea

**Keywords:** osteoporosis, osteoporotic vertebral fracture, genetic factor, epigenetics, DNA methylation, histone modification, non-coding RNA

## Abstract

Osteoporosis is a complex multifactorial condition of the musculoskeletal system. Osteoporosis and osteoporotic vertebral fracture (OVF) are associated with high medical costs and can lead to poor quality of life. Genetic factors are important in determining bone mass and structure, as well as any predisposition for bone degradation and OVF. However, genetic factors are not enough to explain osteoporosis development and OVF occurrence. Epigenetics describes a mechanism for controlling gene expression and cellular processes without altering DNA sequences. The main mechanisms in epigenetics are DNA methylation, histone modifications, and non-coding RNAs (ncRNAs). Recently, alterations in epigenetic mechanisms and their activity have been associated with osteoporosis and OVF. Here, we review emerging evidence that epigenetics contributes to the machinery that can alter DNA structure, gene expression, and cellular differentiation during physiological and pathological bone remodeling. A progressive understanding of normal bone metabolism and the role of epigenetic mechanisms in multifactorial osteopathy can help us better understand the etiology of the disease and convert this information into clinical practice. A deep understanding of these mechanisms will help in properly coordinating future individual treatments of osteoporosis and OVF.

## 1. Introduction

Osteoporosis and osteoporotic fractures are associated with high medical costs and can lead to poor quality of life. Osteoporotic fractures can happen from minor trauma, such as slipping or falling. Worldwide, osteoporotic vertebral fracture (OVF) is the most frequent type of osteoporotic fracture [1,2]. OVF increases in incidence with age and women and is associated with an increased risk of death [2]. Osteoporosis and OVF are influenced by a multifactorial environment, including genetic factors [3,4,5]. Osteoporosis is generally thought to be caused by a reduction in the number of osteoblasts along with preferential differentiation to fat cells in the aged skeleton [6,7]. This may decrease the quantity and functionality of osteoblasts and increase bone marrow fat in aged bones [6,8]. This can result in a reduction in bone formation and bone microarchitecture, leading to additional vertebral fractures and interfering with bone healing and remodeling after fracture [6,9].

Fracture healing reproduces skeletal development and growth with intricate interactions between cells, growth factors, and extracellular matrices. There are usually four stages to fracture repair: (1) inflammation, (2) soft callus formation, (3) hard callus formation, and (4) remodeling (Figure 1) [10]. Each stage is associated with specific cellular and molecular events [11,12,13]. However, in practice, the events are insufficiently described, and there can be extensive redundancies across the different stages. Further, studies over many years have examined the molecular and cellular forces driving the central process [14,15,16]. When considering cellular processes, vascular cells, inflammatory cells, osteochondral precursors, and osteoclasts maintain an important role in the bone repair process. When considering molecular processes, fracture repair is helped by inflammation-promoting cytokines and three main factors: osteogenesis-promoting factor, growth factor, and angiogenesis factor [17]. These factors help set up relevant morphogenetic disciplines by promoting growth or differentiation and recruiting cells. After that, the damaged soft tissue undergoes a repair and the fracture is covered by soft callus and, subsequently, hard callus.

In the final stage of the fracture healing process, remodeling the bone-hardening callus into original cerebellar and cortical structures occurs, and is also known as secondary osteoplasty [13]. This bone remodeling process destroys mineralized bone followed by bone matrix formation that then becomes mineralized. This resorption and formation of bone are highly connected, allowing for skeletal integrity. However, with some pathological conditions such as osteoporosis, bone remodeling steps can uncouple, which can lead to increased fragility and decreased bone strength [18]. To sustain homeostasis of the bone mass, the remodeling of bone integrates extremely regulated steps that rely on the roles and interactions of the osteoblastic and osteoclastic lineages (Figure 2).

Bone cell activation and differentiation are regulated at the molecular level by an intricate signaling network that contains systemic hormones and local secreted molecules. It is well known that many probabilistic and environmental stresses can be modulated through phylogenetic determination and gene expression [19,20,21]. Pathological processes for osteopathy with many factors, such as osteopenia or osteoporosis, can have crucial epigenetic elements. Epigenetic elements can present promising areas of research linking disease risk to genetics and gene expression. New treatment for OVF can be developed in virtue of a greater understanding of epigenetic and molecular regulation of bone cell function and differentiation.

Recently, a new paradigm of epigenetics has advanced. New discipline attempts to identify the delivery of stable phenotypes without altering DNA sequences due to interactions between the genome and internal and external environments. Within epigenetic mechanisms, those that are notable are modifications of histones, DNA methylation, and non-coding RNAs (Figure 3) [22]. Epigenetics can control transcription activity without altering DNA sequences [23] through an enzymatic alteration of 5-cytosine in DNA [24], microRNA [25], histone modifications [26], and remodeling of chromatins. DNA methyltransferases (DNMT) can catalyze methylation of CpG islands of DNA, which can weaken genome stability and reduce the activity of gene transcription [27]. MicroRNA (miRNA) can interrupt mRNA targets and reduce the translation of protein. In addition, it can initiate a post-transcriptional signaling pathway and promote the expression of protein [28,29]. Histone acetyltransferases (HATs) can control histone acetylation, which is necessary to maintain transcription; however, histone deacetylases (HDACs) can eliminate acetyl group from histones, which can favor the formation of heterochromatin to repress the activity of promoters [30].

Histone methyltransferases catalyze the methylation of histone lysine residues to repress transcription of the gene [31]. Histone demethylases can eliminate the methyl group from lysine, which reverses the transcription of the gene [32]. Beyond enzymatic modification, some metabolites, including succinate, butyrate, and propionate, can trigger histone butyrylation, propionylation, succinylation, and crotonylation [33]. A collective review is necessary of the medical role of mRNA, methylated DNA, and histone modification for bone metabolism and osteoporosis occurrence. This article highlights how DNA methylation, histone modification, and non-coding RNA affect osteoporosis, bone remodeling, and treatment of OVF.

## 2. Search Strategy

A systematic search of PubMed, Embase, Web of Science, the Cochrane Database, and KoreaMed was conducted on the 30th of October 2020 independently by two separate reviewers, KKT and YSL. The search terms used were “epigenetics” AND (osteoporosis OR osteoporotic fracture) AND (DNA methylation OR histone modification OR microRNA). Search results were screened by scanning abstracts for the following exclusion criteria: case reports, letters, comments, and papers written in languages other than English. After removing excluded abstracts, full articles were obtained, and studies were screened again more thoroughly using the same exclusion criteria. In addition, we referred to the bibliography of the searched papers. Any discrepancies between the two reviewers were resolved by discussion after the search and a consensus was achieved

## 3. DNA Methylation

The field of bone remodeling makes up a large part of DNA methylation research. Bone remodeling is the main mechanism of the bone healing process for OVF [10]. DNA methylation happens at the 5′ carbon location of the pyrimidine ring of cytosine residue. Previously reported in the context of CpG binucleotide, additional motifs (e.g., C-H-G or C-H-H) are known to be located in embryonic tissue and can induce pluripotent stem cells [34]. Several reports have hypothesized that DNA methylation can play a crucial role in the differentiation of osteoblasts. During osteogenetic differentiation of MSC, changes in promoter methylation can differentiate osteoblasts into an osteoblast runt-associated transcription factor 2 (RUNX2), such that the Osteocalcin gene maintains low methylation of CpG with strong expression. Demethylation of promoters of Osteocalcin, Oserix, and RUNX2 through stopping growth and creating DNA injury (GADD45) has been demonstrated to be associated with osteogenesis differentiation of fat-derived MSCs [35]. In addition, wingless int-1 homolog (Wnt) 3a induces differentiation of osteoblasts only in cells with osteogenic capacity, not in fibroblasts [36]. Wnt3a induces osteoblast differentiation through the stimulation of bone morphogenetic protein 2 (BMP2) production. CpGs island and alkaline phosphatase (ALP) promoters in BMP2 demonstrate increased methylation that leads to preventing their expression in non-osteogenic cells. Moreover, when the non-osteogen cells are exposed to 5dem-aza-2′-deoxycytidine, a CpG demethylating molecule, the BMP2 and ALP genes can receive Wnt3a [36].

Based on initial studies, the crucial signaling molecules in the control of osteoblasts are the bone morphogenetic protein (BMP) and the wingless int-class (Wnt) signaling pathway. Gong et al. showed that dominant-negative homozygous mutation of the LRP5 gene was the cause of osteoporosis pseudoglioma, and Wnt signaling through LRP5 was proved to be closely associated with bone formation [37]. Two later studies demonstrated that LRP5 mutations were associated with hereditary bone mass in humans [37,38,39,40]. Epigenetic changes in the Wnt pathway gene are related to changes in osteoblast function and bone mineral concentration. Wnt inhibitor sclerosis may be mainly expressed in osteoblasts, as it has a paracrine effect on osteoblast function, and it promotes osteogenesis in human and animal models [41,42,43].

Cancellous bone can be affected by osteoporosis earlier than in cortical bone, and a DNA methylation microarray study of the hip cancellous bone in osteoarthritis patients has demonstrated differential methylation [44]. Zhou et al. reported five crucial signaling pathways recently, namely, the signaling pathways of calcium, endocytosis, cyclic guanosine phospho-protein kinase G (cGMPPKG), Rap1, and 5’ adenosine monophosphate-activated protein kinase (AMPK). All were involved in cancellous weakness in postmenopausal women [44].

DNA methylation has also been associated with the value of bone mineral density (BMD). The retinoid X receptor-alpha (RXRA) gene is a necessary cofactor that is connected to the influence vitamin D has on bone. The relationship between RXRA promoter methylation in the umbilical cord DNA of 230 neonates and bone mass at four years of age was demonstrated in a prospectively designed mother–offspring cohort (Southampton Women’s Survey) [45]. Methylation at numerous sites of the promoter of cyclin-dependent kinase inhibitor2A (CDKN2A), which is a gene involved in skeletal development, was correlated with the bone mineral content (BMC), bone surface area, and BMD at four and six years of age [46]. In addition, the methylation level of the endothelial nitric oxide synthase (eNOS) gene promoters in the umbilical cords was strongly correlated with BMC, overall bone surface area, and BMD at nine years of age [47]. In an epigenome-wide study of leukocytes from 5515 European adults, a single CpG location was correlated with BMD of the femoral neck, but this finding was not reproduced in another sample [48]. A different study assessed Alu methylation of leukocytes from 323 postmenopausal women, and Alu hypomethylation was correlated with lower bone mass [49].

## 4. Histone Modification

Histones are very stable proteins that organize, stabilize, and condense DNA within the small area of the nucleus; they encircle genomic DNA around the outer surface. In addition, they geography modify chromatin locally to either increase transcription or decrease it through interactions with DNA packaging proteins. In addition, post-transcriptional modifications, such as acetylation and methylation in the sidechain of amino acids, could control the DNA packaging proteins [50].

Histone acetyltransferase regulation is necessary for bone formation and osteogenic capacity. Further, p300/CBP-associated factor (PCAF) knockdown decreases acetylation at the 9th lysine residue of the histone H3 protein (H3K9ac), which inhibits osteogenic differentiation from mesenchymal stem cells (MSCs) [51].

General control nonderepressible 5 (GCN5) signaling in estrogen deficiency-mediated osteoporosis skeletons is reduced. GCN5 interference weakens the osteoplastic role of mesenchymal progenitors and ectopic osteogenesis [52]. Another study has shown that GCN5 promotes H3K9ac and increases acetyl histone-binding promoters of Wnt signaling proteins of osteogenic precursor cells. Forced expression of GCN5 by the lentivirus shuttle can slow the loss of bone mineral density due to ovariectomy, dynamic osteogenesis histology, morphometry of trabecular bone, and osteogenic differentiation of MSCs [53].

The post-translational methylation of lysine residues of histones modifies the chromatin structure into a heterochromatin that stops active transcription. Several histone methyltransferases and demethylases can catalyze the modification processes. There is more evidence that sheds new light on the epigenetic actions of trimethyl H3K27 (H3K27me3) on tissue deterioration and development. Among all H3K27me3 modifiers, histone methyltransferases polycomb repression complex 2 (PRC2) embryonic ectoderm development (EED), subunits zeste homolog 2 (Ezh2), and SUZ12 catalyze H3K27 trimethylation. Fully transcribed tetratricopeptide repeat X chromosome (UTX) and Jumonji domain containing 3 (Jmjd3) were found to eliminate the trimethyl group of histone [31].

Ezh2 is a histone methyltransferase that suppresses osteoblast maturation and skeletal development. Loss of Ezh2 reduced H3K27me3 levels, increased the expression of osteogenic genes in chondrocytes, and resulted in a transient post-natal bone phenotype. Ezh2 activity is essential for normal chondrocyte maturation and chondral ossification in vivo, although it appears to play a crucial role in the early stages of mesenchymal commitment [54]. Histone methyltransferase Ezh1 and Ezh2 catalyze trimethylation of H3K27 as epigenetic signals of chromatin condensation and transcriptional inhibition. If Ezh1 and Ezh2 are used together in cartilage cells, the skeletal growth of mice is significantly impaired. Both the main processes, cartilage cell proliferation and hypertrophy that form the basis of growth plate cartilage formation, are impaired. A decrease in chondrocyte proliferation is partly due to suppression of the cyclin-dependent kinase inhibitor Ink4ab, while ineffective chondrocyte enlargement is due to the suppression of IGF signaling [55]. Ezh2 loss of pre-confirmation osteoblast due to Cre expression via the OsterixSp7 promoter produces phenotypic normal mice. These Ezh2 conditional knockout mice (Ezh2 cKOs) have normal skulls, collar bones, and long bones, but show increased bone marrow fat and decreased male weight. Ezh2 loss in bone marrow-derived mesenchymal cells suppresses osteogenesis differentiation and inhibits the progression of cell cycles due to a decrease in metabolic activity, a decrease in the number of cells, a change in cell cycle distribution, and the expression of cell cycle markers. Ezh2 plays a bifunctional role by promoting the proliferation of osteogenic cells during osteogenesis and suppressing the commitment of the osteogenesis system [56]. The deletion of Ezh2 in mouse oocytes resulted in a distinct phenotype compared to that resulting from oocyte-specific deletion of EED [57]. H3K36 trimethylation is catalyzed by histone methyltransferase SET-domain-containing 2 (SETD2) to regulate the systematic commitment of bone marrow MSC (BM-MSC). Deletion of Setd2 in mice BM-MSC by conditional Cre expression driven by the Prx1 promoter resulted in bone loss and bone marrow fat increase. Due to the loss of Setd2 in BM-MSCs in vitro, the tendency to differentiate into fat cells rather than osteoblasts was promoted. H3K36 trimethylation mediated by SETD2 can regulate the cell fate of mesenchymal stem cells (MSCs) in vitro and in vivo and can indicate regulation of H3K36 trimethylation levels and/or administration of downstream LBPs by targeting SETD2 [58].

Additionally, the expression of Ezh2 in MSC promotes fat formation in vitro and in vivo and inhibits osteogenic differentiation ability. In contrast, Kdm6a inhibits fat formation in vitro and in vivo and promotes osteogenesis differentiation. Inhibition of Ezh2 activity and knockdown of Ezh2 gene expression in human MSCs resulted in decreased fat formation and increased bone formation. Conversely, knockdown of Kdm6a gene expression in MSC leads to increased fat formation and decreased bone formation [59]. Histone demethylase Kdm4B and Kdm6B play a crucial role in MSC osteogenesis commitment by removing trimethyl groups of H3K9 and H3K27. Exhaustion of Kdm4B or Kdm6B significantly reduces osteogenic differentiation and increases adipogenic differentiation [60]. In addition, Kdm3c had an anti-inflammatory effect on oral bacterial infection through suppression of NF-κB signaling and osteoclastogenesis [61]. A Gene Expression Omnibus (GEO) database of microarray analysis shows that elevation of Kdm5a is confirmed in osteoporosis MSC. Kdm5a inhibits bone formation in an osteoporosis mouse, and pretreatment with a Kdm5a inhibitor partially alleviates bone loss in osteoporosis [62]. UTX weakens glucocorticoid regulation of bone formation and fat formation. UTX reduces RUNX2 promoter methylation and trimethylation of H3K27 concentration in Wnt inhibitor Dickkopf-1 (Dkk1) promoters. β-catenin and Dkk1 regulate glucocorticoid inhibition of UTX signaling. UTX inhibition worsens bone mass, endoplasmic reticulum microstructure, and fatty bone marrow [63].

Acetylation is the most studied section in the clinical application of epigenetics. Adding and removing acetyl groups occurs through histone acetyltransferases and histone deacetylases (Table 1). Since the Sveroranilide hydroxamic acid (SAHA), which is a histone deacetylases (HDAC) inhibitor, was approved by the U.S. Food and Drug Administration (FDA) 10 years ago, SIRT1-7 and other HDACs have attracted attention because of their abnormal histone deacetylase activity in various human diseases such as cancer, viral infection, and neurodegenerative diseases [64]. In total, 18 human HDACs are categorized into four wide classes. HDAC1 selective inhibitors suppress cancer cell proliferation, inflammation, and bone resorption. Both Class I HDAC 8 and Class II HDAC 5 increase during osteoclast development, and both Class I and Class II HDAC inhibit osteoclast absorption in humans [65]. In addition, HDAC activity at least partially regulates osteoblast differentiation and osteogenesis by enhancing RUNX2-dependent transcription activity [66]. The HDAC1 inhibitor promotes bone formation by inducing osteoblast marker genes such as osteopontin and alkaline phosphatase. Consistently, the knockdown of HDAC1 by the short-interference RNA system stimulated osteoblast differentiation [67]. Therefore, the HDAC1 inhibitor is a potentially new class of bone anabolic agents that may be useful in the treatment of osteoporosis. HDAC3 interacts with the amino terminal of RUNX2 and inhibits the RUNX2-mediated activation of the Osteocalcin promoter. HDAC3 regulates the transcription of the osteoblast gene via RUNX2 and promotes the expression of the RUNX2 target gene, osteocalcin, osteopontin, and bone sialoprotein by suppressing HDAC3 [68]. HDAC5 knockout mice exhibit increased mRNA levels (encoded by the SOST gene), more sclerostin-positive osteocytes, reduced Wnt activity, reduced endoplasmic reticulum density, and reduced bone formation by osteoblasts. HDAC5 is regulated in association with the transcriptional activity of the enhancer and suggests direct regulation of SOST gene expression by HDAC5 in bone cells [69]. In contrast, a patient with primary osteoporosis had osteocytes showing increased HDAC5 and low RUNX2 expression [70].

A main transcription factor, NFATc1 (nuclear factor of activated T cell c1), serves to regulate the expression of osteoclast-specific downstream target genes such as TRAP (tartrate-resistant acid phosphatase) and OSCAR (osteoclast-associated receptor). HDAC5 promotes chromatin changes that reduce expression of NFATc1 and excess expression of HDAC5 in bone marrow macrophages and restrict osteoclast differentiation by RANKL [71]. Inhibition of HDAC6 has been shown to reduce the osteoclast activity and modulate gene expression and cytokine production after stimulation with several stimuli [72].

Dynamically, in the absence of a receptor activator for a nuclear factor β-B ligand (RANKL), HDAC7 inhibits osteoclast differentiation by attenuating β-catenin function and cyclin D1 expression, reducing precursor proliferation, and inhibiting NFATc1 and β-catenin-down regulation at the time of RANKL activation [73,74]. Similar to HDAC7, HDAC9 knockout mice have higher numbers of osteoclasts and a lower bone mass [75].

Sirt1 and Sirt6 are the exclusive class III HDACs that have demonstrated a role in bone metabolism and development [76,76]. Sirt1 activation in rat primary bone marrow stromal cells increased the expression of osteoblast markers and also mineralization. In contrast, inhibition of Sirt1 promoted adipogenic differentiation [77]. SIRT6 expression was deregulated during hMSC differentiation. Excessive expression of SIRT 6 was accompanied by reduced expression of specific genes in osteoblast and alkaline phosphatase (ALP) activity. In addition, the TRPV1 channel was also reduced by SIRT6 overexpression through ubiquitinating TRPV1. Capsaicin, which induced overexpression of SIRT6, significantly reduced osteogenesis differentiation [78].

## 5. Noncoding RNA

Noncoding RNA research, particularly in the musculoskeletal area, is still an unexploited field with a new class being updated regularly. Within the non-coding RNA area, microRNA (miRNA) is the most studied category. miRNA is a small non-coding RNA molecule (containing about 22 nucleotides) found in plants, animals, and some viruses, that modulates gene expression. RNA interference is a powerful mechanism for gene silencing, underlying many aspects of eukaryotic biology. At the molecular level, RNA interference is mediated by a ribonucleoprotein complex called RNA-induced silencing complex (RISC) and can be programmed to target virtually all nucleic acid sequences for silencing. RISC’s ability to find target RNA has been adopted many times in evolution, producing a wide range of gene silencing pathways [79]. Several new functions of miRNAs have been proposed, and several novel classes of larger noncoding RNA with a crucial transcriptional role have been recently demonstrated, including small interfering RNA (siRNA), Piwi-interacting RNAs (piRNAs), long non-coding RNAs (lncRNA), and others (Table 2) [80].

### 5.1. MicroRNAs (miRNAs)

miRNAs are small, non-coding RNA molecules. The size of mature miRNAs ranges between 21–23 nucleotides. Their role is to modulate gene expression by post-transcriptional silencing, either by moderating the cleavage of a targeted transcript or by translational inhibition. In addition, miRNA is involved in the epigenetic regulation of bone remodeling (Table 3 and Figure 4).

RUNX2 is a central transcription factor that regulates osteogenesis, particularly osteoblast differentiation, as previously described. The genomic target of RUNX2 is osteoblast-specific cis-acting element 2 (OSE2), an approximately 18-bp, highly conserved DNA sequence located in the promoter region of many osteogenic genes, such as secreted phosphoprotein 1 (SPP1), collagen 1 alpha 1 (COL1A1), alkaline phosphatase (ALP), and others [81]. Beyond the epigenetic controls of RUNX2 transcription, many miRNAs have been shown to be activators or attenuators of its expression. Lineage progression in osteoblast and chondrocyte is strictly controlled by a cell fate-determining transcription factor, RUNX2. During osteogenesis differentiation and cartilage formation, miRNAs (miR-23a, miR-30c, miR-34c, miR-133a, miR-135a, miR-137, miR-204, miR-205, miR-217, miR-218, and miR-338) are generally reversely expressed to RUNX2. All RUNX2 targeting miRNA (except miR-218) significantly inhibits osteoblast differentiation and can reverse its effect by corresponding anti-miRNA [82]. Attenuation of miRNA in protein translation has emerged as a crucial regulator of mesenchymal cell differentiation into osteoblast lines. The deletion of the dicer enzyme in the osteogenesis factor by Col1a1-Cre hindered the survival of the fetus after E14.5. This suggests that dicer-generated miRNA is essential for promoting prenatal osteoblast differentiation and suppressing bone development in adults [83].

Members of the miR-30 family are known as crucial modulators in osteogenic differentiation. miR-26a modulates the expression of the Smad1 protein during the osteoblastic differentiation of human adipose tissue-derived stem cells. Inhibition of miR-26a could increase osteoblast differentiation [84]. miR-135b can control osteoblast differentiation of unrestricted somatic cells (USSCs) by regulating the expression of bone-related genes. A reduction in the expression of osteogenesis markers IBSP and Osterix is known to be involved in bone mineralization in the osteogenesis of USSCs which overexpress miR-135b [85]. Exogenous miR-125b transfection inhibits osteoblast differentiation. In contrast, when endogenous miR-125b is blocked by the transfection of antisense RNA molecules, ALP activity after BMP-4 treatment is increased [86]. Both miR-141 and -200a remarkably modulated BMP-2-induced pre-osteoblast differentiation by the translational repression of Dlx5 [87]. In addition, miR-204/211 acts as a crucial negative regulator of RUNX2, which inhibits bone formation and promotes adipogenic differentiation of mesenchymal progenitor cells and BM-MSCs [88].

A few miRNAs can be promoters of bone formation. miR-218 is induced during osteoblast differentiation and has strong osteogenesis characteristics. miR-218 promotes the involvement and differentiation of bone marrow stromal cells by activating positive Wnt signaling loops [89]. miR-218 expression is promoted by the expression of the Wnt pathway gene Wnt3a, which forms an amplification circuit [90]. MiR-449a inhibits HDAC1 expression and regulates histone acetylation. Accordingly, silencing of endogenous HDAC1 expression by exogenous miR-449a maintains a histone acetylated state, stimulates RUNX2 gene expression, and rapidly promotes osteoblast differentiation [91].

miRNA-directed regulation also targets osteoclasts. Expression of miR-503, targeting RANK, is strongly reduced in circulating CD14+ macrophages of human osteoporosis patients in comparison to normal patients, and it silences miR-503 with an antagomir that promotes RANK expression and osteoclast differentiation, promotes bone resorption, and decreases bone mass in an ovariectomized mouse model [92]. miR-223 is strongly expressed in rheumatoid arthritis (RA) synovial membrane, and excessive expression of miR-223 inhibits osteoclast formation in vitro [93]. In addition, miR-223 expression was significantly higher in the synovial membrane of RA patients and ankle joints of collagen-induced arthritis (CIA) mice than in OA patients and normal mice. The knockdown of miR-233 by lentiviral-mediated silencing reduced the arthritis score, histological score, miR-223 expression, osteoclast formation, and bone erosion in mice with CIA [94].

Several miRNAs have recently drawn scrutiny in the bone field as potential clinical biomarkers for OVF. Ahn identified the TT genotype of miR-149aT>C and suggested it may contribute to decreased susceptibility to OVF in Korean postmenopausal women. The miR-146aCG/miR-196a2TC combined genotype and the miR-146aG/-149T/-196a2C/-449G allele combination may promote increased susceptibility to OVF [4]. In addition, Zarecki et al. showed seven significantly (*p* < 0.05) up-regulated miRNAs (miR-375, miR-532-3p, miR-19b-3p, miR-152-3p, miR-23a-3p, miR-335-5p, miR-21-5p) in patients with OVF [95].

### 5.2. Long Non-Coding (lnc) RNAs

The lncRNAs, large non-coding RNAs > 200 nucleotides in length, play crucial roles in many activities of life [96], such as epigenetic regulation, dose compensation effects, and regulation of cell differentiation (Table 4). Abnormalities of lncRNAs can cause disease, and many reports have demonstrated that lncRNAs are closely associated with the underlying mechanism in the pathogenesis of osteoporosis.

Many lncRNAs have demonstrated an ability to promote osteoblast differentiation, and it has been hypothesized that they could help treat osteoporosis. The histone decarboxylase SIRT1 was shown to be a crucial positive regulator of bone mass and osteoblastogenesis. Nevertheless, the expression of SIRT1 was inversely proportional to lncRNA HIF1A-AS1 expression, which can suggest a role of lncRNA HIF1A-AS1 in osteogenic differentiation [97]. In addition, lncRNA plays a crucial role in gene regulation and is involved in a variety of cellular processes. HoxA-AS3 is increased when inducing adipogenesis of MSC and it interacts with Enhanced Of Zeste 2 (EZH2) and is required for H3K27 trimethylation of the main osteogenesis transcription factor RUNX2. As a result, lncRNA HoxA-AS3 is a crucial molecule in osteoblast differentiation [98]. In contrast to lncRNA HoxA-AS3, lncRNA-differentiation antagonizing non-protein coding RNA (DANCR) was reported to recruit EZH2 in the promotion of H3K27 trimethylation through the interaction with a 305-nt transcript and enhancer of zestehomolog2, which ultimately inhibited transcription of the target gene RUNX2 and osteogenic differentiation [99]. DANCR also modulated the proliferation and osteogenic differentiation of hBM-MSCs via the inactivation of the p38 MAPK signaling pathway [100]. The decline in anti-differentiation noncoding RNA (ANCR) accelerated the growth of periodontal ligament stem cells (PDLSC). Further, the down-regulated ANCR promotes osteogenesis differentiation of PDLSC by up-regulating the osteogenesis differentiation marker gene [101]. In addition, inhibition of lncRNAs, such as HOX transcript antisense RNA (HOTAIR), promoted ALP activity and increased the number of osteogenesis marker genes and calcified nodules in BM-MSC. However, the over-expression of HOTAIR showed the opposite effect. HOTAIR inhibited expression levels of Wntβ-catenin pathway-related proteins [102]. Furthermore, low expression of lncRNAp21 activates the Wntβ-catenin signaling pathway by increasing E2 secretion, ultimately stimulating osteogenesis, and increasing osteogenesis differentiation of MSCs in osteoporosis rat models [103]. The DKK4 gene encodes a protein that belongs to the Dickkopf family. Downregulating lncRNA H19 reduces the expression level of Dkk4, which inhibits the Wnt/β-catenin signaling pathway and negatively regulates osteogenic differentiation [104]. lncRNA AK045490 correlates with osteogenesis differentiation and strengthens skeletal tissue of mice. In the in vitro analysis of BM-MSC, AK045490 inhibited osteoblast differentiation. In vivo inhibition of AK045490 by siRNA saved bone formation in the oophorectomy osteoporosis mouse model. AK045490 inhibits nuclear dislocation of β-catenin and inhibits expression of TCF1, LEF1, and RUNX2 [105]. In a similar fashion, lncRNA AK016739 inhibits osteogenic differentiation and bone formation since it inhibits osteoblastic transcription factors [106]. By inhibiting lncRNA-UCA1, the BMP-2 (Smad158) signaling pathway in osteoblasts is activated to promote osteoblast proliferation and differentiation [107]. Moreover, the expression of serum lncRNA MEG3 in fracture patients is remarkably increased. Because LncRNA MEG3 can promote osteoblast proliferation and differentiation by activating the Wntβ-catenin signaling pathway, it is expected to become a new target for promoting fracture healing [108]. In contrast, downregulating lncRNA MEG3 inhibits osteogenic differentiation by promoting the expression of IGF1 [109]. The considerations outlined here suggest that many lncRNAs have inhibitory effects on osteogenic differentiation. As a result, silencing the expression of these specific lncRNAs using specifically targeted drugs may be possible, which may limit osteoporosis development.

lncRNAs have been reported to modulate osteoclastogenesis by modulating the expression of specific target mRNAs. Plasma lncRNA TUG1 was regulated at a higher level in patients with osteoporosis than in healthy participants. A patient with osteoporosis and a healthy patient is distinguished by increasing the regulation of plasma lncRNA TUG1. LncRNA TUG1 levels increased with advances in the clinical stage. Although excessive expression of lncRNA TUG1 accelerated proliferation and inhibited apoptosis of mouse osteoclasts, lncRNA TUG1 siRNA silencing played a reverse role [110]. The expression of lncRNA AK077216 is remarkably suppressed during osteoclast formation. Up- and down-regulation of lncRNA AK077216 promotes and inhibits osteoclast differentiation, bone absorption, and expression of related genes. lncRNA AK077216 regulates the expression of NFATc1 and promotes osteoclast formation and function [111]. Downregulation of lncRNA SNHG15 inhibits osteoclasts by modulating the RANK/RANKL pathway [112]. lncRNAs also have inhibitory effects on osteoclasts. These lncRNA expression levels were inversely correlated with osteoporosis severity. lncRNA RP11-498C9.17 is strongly correlated with many epigenetic regulatory factors, including HDAC4, HMGA1, MORF4L1, and DND1. Downregulation of HDA was reported to increase osteoclast differentiation, which suggests that lncRNA RP11-498C9.17 may modulate osteoclast production via HDAC4 signaling [113]. In addition, lncRNA Bmncr inhibits RANKL-induced osteoclast differentiation [114]. Upregulation of lncRNA NONMMUT037835.2 inhibits osteoclast differentiation, and downregulation of lncRNA-NONMMUT037835.2 promotes osteoclast formation [115]. In conclusion, these lncRNAs with effects on osteoclast differentiation may demonstrate a breakthrough in treating osteoporosis. The expression level of lncRNAs promoting osteoclast differentiation is lowered and suppressed.

## 6. The Possibility of Epigenetics in Treatment of Osteoporosis and OVF

The epigenetic effects on bone formation and remodeling reactions may facilitate the development of epigenetic therapeutics with the potential to treat osteoporosis and OVF. A large variety of proof-of-concept studies have demonstrated the remedial effects of miRNA and epigenetic modifiers in slowing osteoporosis development. A recombinant adeno-associated virus serotype 9 (rAAV9) can deliver artificial miRNA (amiR) to osteoclast cells of patients/animals with osteoporosis to silence expression of RANK and cathepsin K (rAAV9.amiR-rank, rAAV9.amiR-ctsk), which are major osteoporosis regulators. Because rAAV9 is very effective for the transduction of osteoclasts, systemic administration of rAAV9 with amiR rank or amiR-ctsk results in a significant increase in bone mass in mice. rAAV9.amiR-ctsk suppresses bone loss and improves bone mechanical properties in postmenopausal and senile osteoporosis mouse models [116]. A circulating miR-338 cluster in the serum could maintain bone formation capacity and increased bone mass and trabecular structure in an osteoporosis mouse model [117]. miR-672-5p induced osteoblast differentiation and mineralization in ovariectomized mice [118]. Histone methyltransferase DOT1L inhibition decreases osteoclastic activity, which can delay osteoporosis progression [119]. The knockdown of EZH2 by lentivirus-expressing shRNA rescued the abnormal fate of osteoporotic MSC. The H3K27me3 inhibitor DZNep effectively derepressed Wnt signaling and improved osteogenic differentiation of osteoporotic MSCs in vitro [120]. DNA methylation inhibitor 5-aza-20-deoxycytidine improves bone mass in disuse-induced osteopenic bone development [121].

## 7. Future Perspectives

Epigenetics, a genetic change in gene expression unrelated to fundamental changes in the genetic code, has emerged as a crucial and promising field of research in the larger field of bone remodeling. Epigenetic changes in central genes associated with osteoblast, osteoclast differentiation, and cell signaling are closely related to bone remodeling and healing, in healthy tissues and in diseases of dysregulated remodeling, such as osteoporosis and OVF. In fact, epigenetic mechanisms can be isolated and targeted by pharmacological agents, an approach that is effective for specific tumors and neurological disorders, so future treatment of bone disease is highly anticipated. Precision medicine allows you to use all of these features to tailor your treatment to individuals. Future research in this field will certainly demonstrate further discovery of the underlying mechanisms of bone biology, along with the balance between metabolism and catabolism in bone tissue. It provides new diagnostic and therapeutic possibilities for patients with osteoporosis and OVF.

## 8. Conclusions

In recent years, an abundance of new epigenetic data has been accumulated in the fields of osteoporosis and OVF. Several epigenetic biomarkers and, in some cases, molecular signatures have been identified, forming potential therapeutic or diagnostic targets. This review illustrates that epigenetic regulation is deeply involved in bone physiology and that epigenetic-based therapy and diagnosis have already shown potential in the fields of osteoporosis and OVF.

## Figures and Tables

**Figure 1 ijms-21-09455-f001:**
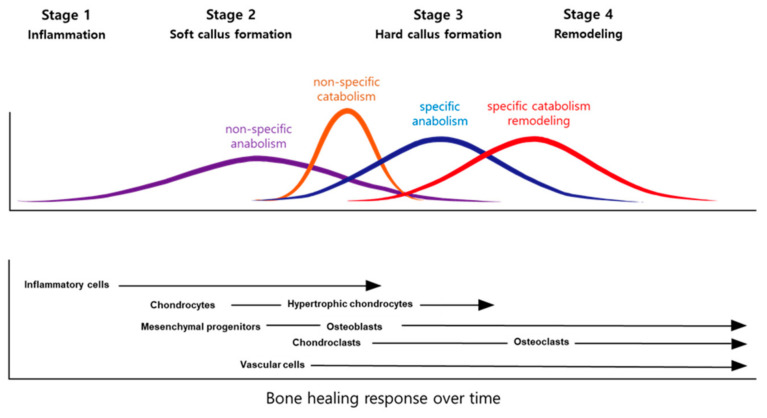
Models of the fracture healing process.

**Figure 2 ijms-21-09455-f002:**
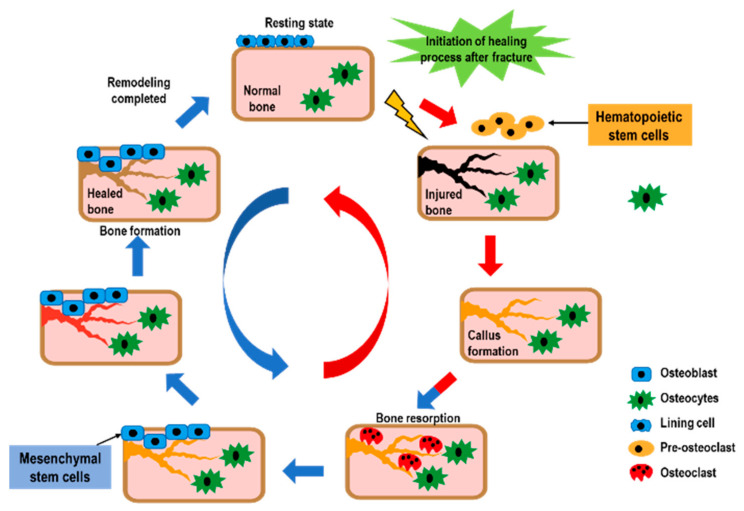
The bone healing process after a fracture. Bone remodeling after fracture is initiated by osteoclast activation, and it solubilizes the bone mineral and degrades the matrix. Osteoclasts originate from hematopoietic stem cells which fuse to form multinucleated cells (activated form of osteoclasts). Monocytes/macrophages remove debris, followed by a bone formation phase performed by osteoblasts, producing osteoid matrix which will mineralize.

**Figure 3 ijms-21-09455-f003:**
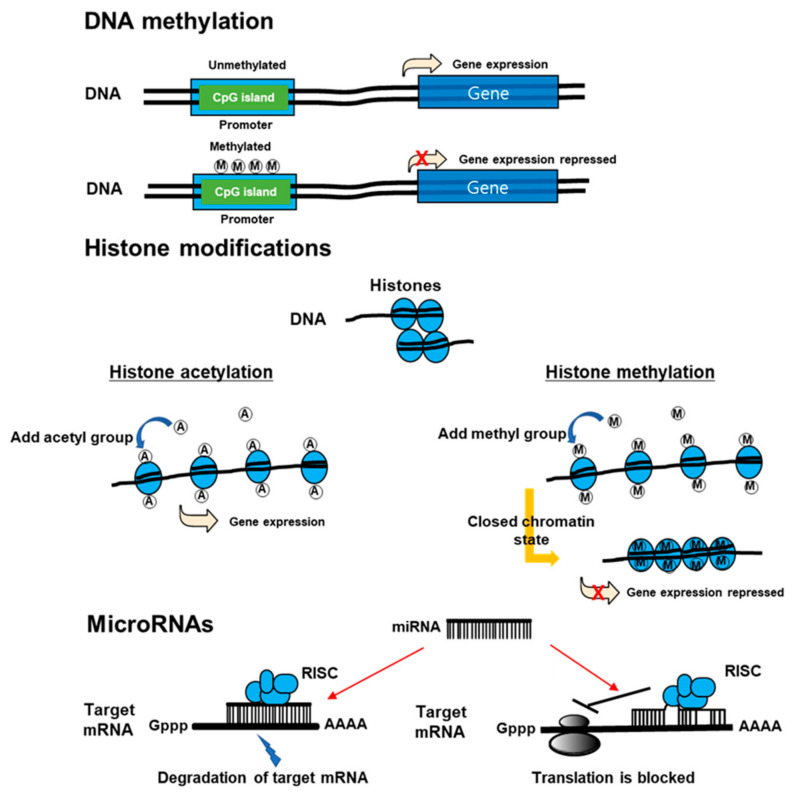
The mechanisms of epigenetics. DNA methylation inhibits DNA transcription. Histone modification, which modifies the interaction of DNA with histones or other DNA-binding protein complexes, regulates gene expression in a more dynamic and changing manner than does DNA methylation. Non-coding RNAs may be large (>200 nucleotides) or small (18–200 nucleotides). mRNA is one of the small non-coding RNAs. miRNAs suppress gene expression by selectively binding to the 3′ non-coding region (3′UTR) of their mRNA targets through base-paring. miRNAs can negatively regulate gene expression by two different post-transcriptional mechanisms. RISC: RNA induced silencing complex.

**Figure 4 ijms-21-09455-f004:**
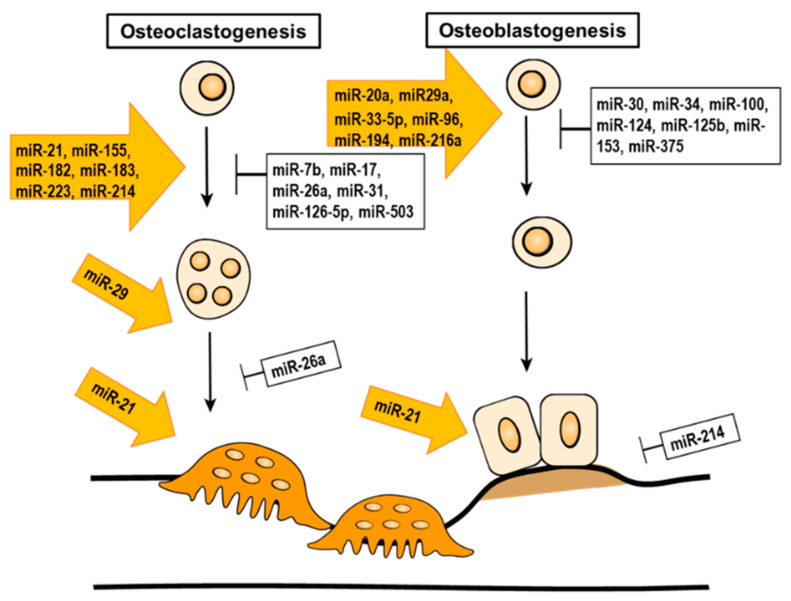
The epigenetic regulation of miRNA in bone remodeling.

**Table 1 ijms-21-09455-t001:** The role of histone deacetylase (HDACs) in bone.

HDAC	Class	Mechanism	Functions in the Bone
HDAC1	I	Down-regulation of RUNX2	Suppress the differentiation of osteoblasts
HDAC2	I	Down-regulation of FoxO1	Promote RANKL-induced osteoclastogenesis
HDAC3	I	Down-regulation of RUNX2	Maintain bone mass
HDAC4	II	Down regulation of RUNX2	Suppress endochondral ossification
HDAC5	II	Down regulation of RUNX2	Suppress the differentiation of osteoblasts
HDAC7	II	Down regulation of RUNX2	Regulate endochondral ossification
HDAC8	I	Up-regulation of Homeobox transcription factors Otx2 and Lhx1	Regulate the intramembranous ossification
HDAC9	II	Down-regulation of RANKL	Suppress osteoclastogenesis

RUNX2 (runt-related transcription factor 2), FoxO1 (forkhead box protein O1), Otx2 (orthodenticle homeobox 2), Lhx1 (LIM/homeobox protein Lhx1), RANKL (receptor activator of nuclear factor kappa-Β ligand).

**Table 2 ijms-21-09455-t002:** The various non-coding RNAs (ncRNA) and their characteristics.

ncRNA	Length (nt)	Characteristics
MicroRNA(miRNA)	20–24	- Single-stranded RNA derived from pre-miRNA - Silencing of genes
Small interfering RNA (siRNA)	20–24	- Double-stranded RNA processed by endoribonuclease Dicer into mature siRNA - Protection against viral infection - Post-transcriptional silencing/RNA interference
Piwi-interacting RNA (piRNA)	24–31	- Make the complexes with P-element induce wimpy testis (Piwi) proteins of the Argonaute family - Silencing of transposable elements
Promoter-associated RNA (PAR)	16–200	- Single-stranded RNA with half-life - Regulation of post-transcription
Enhancer RNA (eRNA)	100–9000	- Single-stranded RNA with half-life - Activation of transcriptional genes
Long non-coding RNA (lncRNA)	>200	- Non-protein coding transcripts - Modifications of post-transcription - Transcriptional/post-transcriptional regulation - Precursor of siRNA

**Table 3 ijms-21-09455-t003:** The pathway/affected molecule of miRNAs and their effects on bone biology.

miRNA Family	Pathway or Affected Molecule	Effect on Osteoblasts and Osteoclasts
miR-33-5p	Hmga2	Promote the differentiation of osteoblasts
miR-96	EGFR, HB-EGF Wnt/β-catenin signaling pathway	Promote the differentiation of osteoblasts
miR-139-5p	NOTCH1, Wnt/β-catenin pathway	Promote the differentiation of osteoblasts
miR-194	RUNX2	Promote the differentiation of osteoblasts
miR-216a	PI3K/AKT pathway BMP/TGF-β signaling pathway	Promote the differentiation of osteoblasts
miRNA-433-3p	DKK1	Promote the differentiation of osteoblasts
miR-542-3p	SFRP1 BMP-7/PI3K-surviving pathway NKIRAS2, NF-κB signaling pathway	Promote the differentiation of osteoblasts Inhibit the differentiation of osteoblasts
miR-26a	Smad1	Inhibit the differentiation of osteoblasts
miR-100	Smad1	Inhibit the differentiation of osteoblasts
miR-124	Dlx3, Dlx5, and Dlx2 GSK-3β, Wnt/β-catenin pathway	Inhibit the differentiation of osteoblasts
miR-125b	BMPR1b	Inhibit the differentiation of osteoblasts
miR-153	BMPR2	Inhibit the differentiation of osteoblasts
miR-203a-3p	Smad9, Wnt/β-catenin signaling pathway	Inhibit the differentiation of osteoblasts
miR-214-3p	ATF4	Inhibit the differentiation of osteoblasts
miR-375	RUNX2	Inhibit the differentiation of osteoblasts
miR-19a	TWIST and RUNX2	Promote the differentiation of osteoclasts
miR-21	RANKL, PI3K/Akt signaling pathway, PDCD4	Promote the differentiation of osteoclasts
miR-155	TNF-α, IL-1β, RANKL, M-CSF, RANK, TRAP, Bcl-2, LEPR, AMPK, p-AMPK, OPG, Bax, TAB 1	Promote the differentiation of osteoclasts
miR-182	Foxo3, Maml1	Promote the differentiation of osteoclasts
miR-183	RANKL, HO-1	Promote the differentiation of osteoclasts
miR-223	TWIST and RUNX2	Promote the differentiation of osteoclasts
miR-214	Pten, PI3K/Akt pathway	Promote the differentiation of osteoclasts
miR-7b	DC-STAMP	Inhibit the differentiation of osteoclasts
miR-17	RANKL	Inhibit the differentiation of osteoclasts
miR-26a	CTGF/CCN2	Inhibit the differentiation of osteoclasts
miR-31	RhoA	Inhibit the differentiation of osteoclasts
miR-126-5p	PTHrP and MMP-13	Inhibit the differentiation of osteoclasts
miR-141	Calcr, EphA2	Inhibit the differentiation of osteoclasts
miR-503	RANK	Inhibit the differentiation of osteoclasts

Hmga2 (high-mobility group AT-hook 2), EGFR (epidermal growth factor receptor), HB-EGF (heparin-binding EGF-like growth factor), NOTCH1 (notch homolog 1, translocation-associated (Drosophila)), RUNX (runt-related transcription factor), PI3K (phosphatidylinositol 3-kinase), AKT (protein kinase B), BMP (bone morphogenetic protein), TGF-β (transforming growth factor beta), DKK1 (Dickkopf Wnt signaling pathway inhibitor 1), SFRP1 (secreted frizzled-related protein 1), NKIRAS2 (NFKB Inhibitor Interacting Ras Like 2), Dlx (Distal-Less Homeobox), GSK-3β (Glycogen synthase kinase 3β), BMPR (bone morphogenetic protein receptor), ATF4 (activating transcription factor 4), TWIST (Twist-related protein), RANKL (receptor activator of nuclear factor kappa-Β ligand), PDCD4 (Programmed cell death protein 4), TNF (tumor necrosis factor), IL (interleukin), M-CSF (macrophage colony-stimulating factor), TRAP (tartrate-resistant acid phosphatase), Bcl-2 (B-cell lymphoma), LEPR (leptin receptor), AMPK (5′ adenosine monophosphate-activated protein kinase), OPG (Osteoprotegerin), Bax (Bcl-2-associated X protein), TAB 1 (TGF-beta activated kinase 1 binding protein 1), Foxo3 (forkhead box O3), Maml1 (mastermind-like protein 1), HO-1 (heme oxygenase-1), Pten (phosphatase and tensin homolog), DC-STAMP (dendritic cell-specific transmembrane protein), CTGF/CCN (connective tissue growth factor), RhoA (ras homolog gene family, member A), PTHrP (parathyroid hormone-related peptide), MMP-13 (matrix metallopeptidase 13), Calcr (calcitonin receptor), EphA2 (ephrin type-A receptor 2).

**Table 4 ijms-21-09455-t004:** The pathway/affected molecule or lncRNAs and their effects in bone biology.

lncRNA	Pathway or Affected Molecule	Effect on Osteoblasts and Osteoclasts
lncRNA HIF1A-AS1	SIRT1	Promote the differentiation of osteoblasts
LncRNA HoxA-AS3	EZH2, H3K27me3, RUNX2	Promote the differentiation of osteoblasts
lncRNA MALAT1	miR-143, miR-204	Promote the differentiation of osteoblasts
lncRNA MODR	miR-454	Promote the differentiation of osteoblasts
LncRNA KCNQ1OT1	miR-214	Promote the differentiation of osteoblasts
LncRNA NTF3-5	miR-93-3p	Promote the differentiation of osteoblasts
LncRNA POIR	miR-182	Promote the differentiation of osteoblasts
LncRNA Linc-ROR	miR-145	Promote the differentiation of osteoblasts
LncRNA H19	miR-675, miR-141, miR-22 Wnt/β-catenin pathway	Promote the differentiation of osteoblasts Inhibit the differentiation of osteoblasts
LncRNA-DANCR	EZH2, H3K27me3, RUNX2, p38 MAPK	Inhibit the differentiation of osteoblasts
LncRNA ANCR	Wnt/β-catenin pathway	Inhibit the differentiation of osteoblasts
LncRNA HOTAIR	Wnt/β-catenin pathway	Inhibit the differentiation of osteoblasts
lncRNA p21	E2, Wnt/β-catenin pathway	Inhibit the differentiation of osteoblasts
Lnc-AK045490	β-catenin, TCF1, LEF1, and RUNX2	Inhibit the differentiation of osteoblasts
Lnc-AK016739	osteoblastic TF	Inhibit the differentiation of osteoblasts
lncRNA UCA1	BMP-2/(Smad1//5/8)	Inhibit the differentiation of osteoblasts
LncRNA MEG3	Wnt/β-catenin signaling pathway IGF1	Promote the differentiation of osteoclasts Inhibit the differentiation of osteoblasts
LncRNA HOTAIR	miR-17-5p, Smad7	Inhibit the differentiation of osteoblasts
LncRNA MIAT	miR-150-5p	Inhibit the differentiation of osteoblasts
lncRNA-ORLNC1	miR-296	Inhibit the differentiation of osteoblasts
LncRNA MEG3	miR-133a-3p	Inhibit the differentiation of osteoblasts
LncRNA TSIX	miR-30a-5p, and RUNX2	Promote the apoptosis of osteoblasts
lncRNA TUG1	PTEN	Promote the differentiation of osteoclasts
lncRNA AK077216	NIP45, NFATc1	Promote the differentiation of osteoclasts
lncRNA SNHG15	RANK/RANKL pathway	Promote the differentiation of osteoclasts
LncRNA-Jak3	NFATc1, CTSK	Promote the differentiation of osteoclasts
LncRNA LINC00311	DDL3	Promote the differentiation of osteoclasts
LncRNA RP11-498C9.17	HDAC4	Inhibit the differentiation of osteoclasts
LncRNA Bmncr	RANK	Inhibit the differentiation of osteoclasts
LncRNA NONMMUT037835.2	RANK, NF-κB/MAPK signaling pathway	Inhibit the differentiation of osteoclasts
LncRNA-NEF	IL-6	Inhibit the differentiation of osteoclasts

HIF1A-AS (Hypoxia-inducible factor-1A-antisense RNA), SIRT (Sirtuin), EZH2 (enhancer of zeste homolog 2), H3K27me3 (polycomb repressive complex 2 methylates lysine 27 of histone H3-mediated trimethylation), RUNX2 (runt-related transcription factor 2), MAPK (mitogen-activated protein kinas), TCF1 (transcription factor 1), LEF1 (lymphoid enhancer-binding factor 1), TF (transcription factor), BMP (bone morphogenic protein), IGF (insulin-like growth factor), PTEN (phosphatase and tensin homolog), NIP45 (Nuclear Factor Of Activated T Cells (NFAT) 1interacting protein), RANKL (receptor activator of nuclear factor kappa-Β ligand), CTSK (cathepsin), DDL3 (delta-like 3), HDAC4 (histone deacetylase 4), NF-κB (nuclear factor-κB), MAPK (mitogen-activated protein kinase), IL (interleukin).

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
