# Peer review of "The Role of Epigenomics in Osteoporosis and Osteoporotic Vertebral Fracture"

_ijms, 2020, doi:10.3390/ijms21249455_

Round 1

Reviewer 1 Report

This is a well written and interesting review on the role of epigenetic regulations of gene expression in bone metabolism, both at healthy and pathological levels. Figures and tables are well done and very useful.

Just few minor points:

  1. Authors reported, along the text, RUNX2 or Runx2. Please check the use of capital letters. Usually only the first one capital letter is used for mice gene or protein. Human gene is
  2. Page 3 line 81. Please remove “Table 1” since this table is referred only to HDACs involved in bone metabolism and should be not cited at this point of the text.
  3. Caption of Figure 3, line 104. the term “mRNA” should be “miRNA”
  4. Page 6, line 205. The “in vitro and in vitro” is wrong and it should be replaced by “in vitro and in vivo”
  5. Table 2. In the fourth line of the table there is a wrong bracket after the term “piRNA”
  6. Page 9, line 285. Please move the terms “Table 3” from the line 285 to the line 286 after the “Figure 4” terms
  7. Page 10, line 298. The term “differentiation” should be removed.
  8. Page 14, lines 414-415. Please rewrite the sentence: “ A recombinant adeno-associated virus serotype 9 (rAAV9) can deliver artificial miRNA (amiR) to osteoporosis”. Osteoporosis is not a type of cell you can deliver miRNA to. I suppose it should be “can deliver artificial miRNA (amiR) to osteoclast cells of patients/animals with osteoporosis”
  9. Page 14. Lines 430-431. What do the Authors mean with this sentence? Please rewrite, since it is unclear.
  10. A paper by Luzi E et al. “The regulatory network menin-microRNA 26a as a possible target for RNA-based therapy of bone diseases” (Nucleic Acid Ther 2012 Apr;22(2):103-8) described the role of this miRNA in osteogenic differentiation of mesenchymal progenitors. It should be cited in the text, and miR-26 should be added in Table 3.

Author Response

Dear Reviewer 

I appreciate your excellent review. Thank you so much.

Please find my attached file. 

Thanks, again. 

Reviewer 2 Report

This is a hugely interesting paper that discusses the epigenetics contribution via several mechanisms during physiological and pathological bone remodeling. The authors included an adequate number of figures (4) and tables (4), making the paper more clear, useful, and practical. It provides information and inspires thoughts in the researched area. It was a pleasure to read, and the readership is going to benefit.

Minor Comments:

The current manuscript looks like a textbook. You could add some information about the methodology followed, even this is not a systematic review.

I would recommend a separate “future perspectives” and “conclusions” section to make your points more clear.

Author Response

Dear Reviewer. 

I appreciate your excellent comments and suggestions. 

Please find my attatched file. 
